# Foodborne Botulism in Western Romania: Ten Years’ Experience at a Tertiary Infectious Disease Hospital

**DOI:** 10.3390/healthcare9091149

**Published:** 2021-09-02

**Authors:** Iosif Marincu, Felix Bratosin, Iulia Vidican, Bianca Cerbu, Oana Suciu, Mirela Turaiche, Livius Tirnea, Madalina Timircan

**Affiliations:** 1Methodological and Infectious Diseases Research Center, Department of Infectious Diseases, “Victor Babes” University of Medicine and Pharmacy, 300041 Timisoara, Romania; imarincu@umft.ro (I.M.); iulia.georgianabogdan@gmail.com (I.V.); ionitabiancaelena@yahoo.com (B.C.); oana_suciu96@yahoo.com (O.S.); mirela.turaiche@gmail.com (M.T.); liviustirnea@yahoo.com (L.T.); 2Department of Gynecology, “Victor Babes” University of Medicine and Pharmacy, 300041 Timisoara, Romania; timircan.madalina@yahoo.com

**Keywords:** botulism, botulinum neurotoxin, foodborne botulism, food poisoning

## Abstract

Objectives: The purpose of this study was to analyze epidemiological data concerning foodborne botulism in Western Romania over the last decade. Botulism, the toxin formed by the bacterium *Clostridium botulinum*, results in a neuroparalytic disorder capable of severe clinical progression that begins in the cranial nerves and progressively descends. Preventing progression to a severe case entails timely diagnosis since curative assets are restricted. Ingesting food containing a preformed toxin (foodborne botulism) is the most typical form. Methods: Medical records were retrospectively analyzed from 2010 to 2020 for all food botulism cases. A seroneutralization test was performed with type A, B and E anti-botulinum sera to establish the kind of toxin involved. Results: Overall, 18 cases of foodborne botulism were admitted to the hospital during this period and confirmed by laboratory analysis. Most of the participants in our study were men (61.1%), and 77.8% of the total lived in rural areas. All the participants showed classic symptoms of botulism, and dysphagia was present in all cases. The trivalent ABE antitoxin was administered by the hospital, and toxin type B was isolated in all patients. The main sources of the toxin were pork, ham and canned pork meat. Conclusions: Stronger efforts are needed to foster community awareness of foodborne botulism, particularly in home-preserved food.

## 1. Introduction

Foodborne botulism is an unusual illness with an unpredictable progression that is closely correlated with food canning traditions and certain product handling techniques [1,2,3,4,5,6]. The disease mostly affects humans but can be found in animals. The illness was initially observed in the 17th century in Europe specifically linked to sausages. However, the first complete description of clinical symptoms was published between 1817 and 1822 by the German physician and poet Justinus Kerner (1786–1862), who nicknamed the toxin “sausage poison botulus” based on the Latin “botulus” [7]. The disease was known in Europe during the late 18th century and later was accepted as a foodborne disease throughout the northern hemisphere [8]. A total of eight *Clostridium botulinum* neurotoxins (BoNTs) were identified: A, B, C, D, E, F, G, and X [9], and each type can yield more than 40 subtypes [10]. Food poisoning in humans is a clinical feature of neurotoxins A, B, and E [11], while C and D affect animals although consuming fish products may cause type E botulism, according to some reports [12]. Regarding its properties, the toxin has no aftertaste or smell, so it does not change the taste or smell of food. After consuming BoNT-contaminated food, the preformed toxin is quickly absorbed and accumulates in synaptic clefts at neuromuscular junctions where it blocks the release of acetylcholine into motor nerve terminals. Hence, the nerve inflow conduction is blocked, causing a two-sided and equal paralysis of the eye muscles and pharynx. The muscle paralysis involving the eyes commonly fades within 4 to 6 weeks since new synaptic knobs are formed to replace those affected by the toxin [1]. The diaphragm is a vital muscle that can also be affected by BoNT. Blocking the acetylcholine-induced contraction triggered at phrenic nerve endings can cause respiratory failure that necessitates rapid medical attention and intensive care, but most of the time the patient does not survive.

The incubation phase of foodborne botulism is 12–48 h after ingestion. The initial signs of illness are ptosis, diplopia, and dehydration of the oral and pharyngeal mucosa, followed by flaccid laryngeal and pharyngeal muscle paralysis. Gastrointestinal involvement may lead to diarrhea, emesis, nausea, and abdominal pain, but the botulism toxidrome varies depending on the type, patient, and amount of toxin consumed [13]. Because these symptoms are not disease-specific, changes in appearance can make diagnosis challenging. A definite diagnosis is reinforced by the indications mentioned above and confirmed by assaying botulinum toxin in serum. The core treatment is thorough: intensive care provision, ventilatory support if necessary, and antitoxin provision. Well-timed antitoxin management may arrest the progress of paralysis and shorten the disease course [14]. Complete recovery usually takes weeks to months, aided by prompt administration of trivalent botulinum antitoxin serum and respiratory support if needed [5,6].

Romania has faced numerous sudden foodborne botulism outbreaks over a short time: In 2003, 27 cases including two deaths; in 2004, 18 cases over four months; in 2005, 21 cases in three outbreaks; in 2006, 23 cases and one death in two outbreaks; in 2007, 110 cases with three deaths in five outbreaks nationwide; and in 2008, 11 cases in one outbreak [15]. In these outbreaks, 98.75% of patients ingested B-type toxin; E-type toxin accounted for the remaining 1.25% [16]. In 2018, there were 24 suspected cases of type B botulism, of which 15 were confirmed, and one was rated probable. The study was designed to explore retrospectively the clinical features and final laboratory diagnoses of botulism cases admitted to Victor Babes Hospital of Infectious Diseases over 10 years (2010–2020), to address the question of whether botulism is a significant healthcare awareness issue in Romania.

## 2. Materials and Methods

The medical records of all participants with foodborne botulism were retrospectively analyzed from 1 January 2010 to 31 December 2019 after being admitted to the Department of Infectious Diseases, Clinical Hospital for Infectious Diseases and Pneumophysiology “Dr. Victor Babes”, which serves three counties in Western Romania. All patients gave informed written consent to allow personal data collection, which included demographic data and laboratory analyses collected from an electronic database and paper charts. Descriptive statistics were used, in which the mean, standard deviation, and frequency were calculated for continuous variables. The data were analyzed by SPSS version 26.

The botulinum neurotoxin was identified in serum samples taken from study participants on the first day of admission immediately prior to beginning treatment with anti-botulinic serum. Within the first 48 h of patient admission, all cases were reported as required to the local public health department, which ordered an epidemiological investigation into these incidents.

The Mouse Lethality Assay (MLA) is the conventional method for identifying the presence of BoNT and evaluating its biologic activity, and it was used in the current research following the “AOAC Official Method 977.26” [17]. Mice were given intraperitoneal inoculations with the minimum lethal dose to confirm the clinical diagnosis of botulinum toxins in the blood samples. The seroneutralization test was performed with type A, B and E anti-botulinum sera (bioMérieux, Marcy l’Etoile, France) to establish the botulinum toxin subtype. The laboratory tests were executed at the National Institute of Research and Development for Microbiology and Immunology, “Cantacuzino” (Bucharest, Romania). Ethical approval was acquired from the Committee of Ethics of the Victor Babes Hospital in Timisoara, Romania, with approval number 601.

## 3. Results

From 2010 to 2020, 18 patients with foodborne botulism were admitted to Victor Babes Hospital. The majority exhibited gastrointestinal symptoms within 12–24 h after eating contaminated food. Dysphagia was present in all patients, along with headache and dizziness, 88.9 and 83.3% of cases, respectively (Table 1). Other symptoms were double vision (*n* = 12), constipation (*n* = 12), ptosis (*n* = 12), fatigue (*n* = 11) and insomnia (*n* = 11). Table 1 shows the classic symptoms of the participants.

One patient, who developed complications from pneumonia and needed urgent intubation and ventilatory support, was moved to the Intensive Care Unit. Additionally, six (33.3%) patients had isolated cases of foodborne botulism where an outbreak was mostly within the family. They were all treated with trivalent ABE antitoxin in hospital. The most common comorbid condition was hypertension, followed by diabetes mellitus and gout. All patients had a favorable outcome and recovered with no medical sequelae.

All cases related to botulism by years are described in Figure 1, and case distribution by province in Figure 2. The mean age of participants was 48 years, ranging from 19 to 72 years. Fourteen cases (77.8%) came from rural areas, and four (22.2%) lived in urban regions. Men accounted for 11 of the cases (61.1%). The mean incubation phase was 30 h, ranging from 12 h to 8 days. Co-existing medical conditions and the acute presentation of foodborne botulism were seen in six of the participants (33.3%). After injecting the trivalent anti-botulinum toxin, seven patients (38.9%) developed an allergic reaction. The demographic details are presented in Table 2.

The food usually associated with botulism is old-style preserved pork. In nine cases (50%), pork ham caused botulism; home-canned pork and canned meat were observed in eight cases (44.4%); and vacuum-packed pâté in one (5.6%). Food portions were not examined but determined based on possible inadequate decontamination in addition to the food’s sensorial properties. Serum samples for toxin isolation were gathered before the injection of the botulinum antitoxin dose. The type of toxin was recognized by MLB and organized by toxin removal using the homogenous monovalent-type precise antitoxin.

Of the 18 laboratory-confirmed cases, *Clostridium botulinum* type B toxin was detected in all the patients’ sera. Stool analysis to separate neurotoxin types was not conducted for any of them, which is a drawback of our study.

## 4. Discussion

Our study specified an analysis of the electronic records of cases of botulism in western Romania admitted to Victor Babes Hospital of Infectious Diseases in Timisoara over 10 years (2010–2020) following significant radical and financial alterations. In previous studies, the prevalence of botulism from 1980 to 1989 for Romania was 0.05 per 100,000 individuals annually. From 1990 to 1998 the rate rose to 0.06, and from 1999 to 2006 it was 0.09 per 100,000 individuals [4]. It is believed that the higher incidence of foodborne botulism in the 1990s, was linked to an increase in poverty following the fall of the communist era in 1989, after which people were inclined towards food preservation. In one of the studies, the data showed that botulism throughout Romania was a serious problem; however, the condition in the southwest was quite encouraging as there was a decrease in the number of cases and a prevalence, that from 1999 to 2007, fell to 0.05 per 100,000. Our study showed decreased prevalence each year. The highest number of botulism cases occurred in 2015. The decreasing trend reflected the effectiveness of healthcare strategies and public mindfulness in this area. The minimal occurrence of botulism in this region could be due to less preserved food usage, and the public adoption of safe procedures for meat and seafood consumption.

According to our data, botulism was more common in men (61.1%) than women, and most cases were in western rural areas. These findings were similar to surveillance studies conducted in Italy from 1986 to 2015 [3]. These stated that confirmed male cases were 51.7%, and most patients came from the countryside in the southern and central districts. In 2014, another European study confirmed 123 cases, of which 67 were male patients [18]. 

The neurological symptoms caused by BoNT require an accurate, conclusive diagnosis to dismiss other probable nervous conditions, such as myasthenia gravis, Guillain–Barre syndrome, cerebrovascular events, or other types of food poisoning. In our study, the most typical symptom was dysphagia, which was seen in all cases. Furthermore, there was frequent monitoring of respiration, perfusion and upper airway integrity, as well as continuous pulse oximetry, spirometry, and arterial blood gas measurement. Orotracheal intubation was considered for patients with a compromised upper airway, and if vital respiratory capacity fell below 30% of the predicted value, ventilatory support was offered to compensate for paralyzed respiratory muscles.

This was seen in one of our patients, who developed pneumonia and required ventilatory assistance. According to international surveillance statistics, the mortality rate is lower when all affected patients are given the corresponding antitoxin. The mortality rate in developed countries for individually treated patients is approximately 5% to 10% [19].

Home handling of traditional pork products without insufficient sanitary knowledge subjects consumers to several hazards, depending on cooking style and community influences. As in other parts of the world [20], the most common infection hazard is *Salmonella spp*. and *Trichinella spiralis* infection. On rare occasions, contamination with BoNT occurs, or *Streptococcus suis* is involved in sporadic cases of purulent meningitis, sepsis, uveitis, and arthritis, which affects people with a history of occupational hazards or who come into close contact with infected pigs [21]. In Romania, domestic pigs are slaughtered in the winter, and the meat is preserved, canned, and kept at room temperature for a few months before ingestion. *C. botulinum* neurotoxin types B, E, and F have been seen to grow at the low temperatures used for refrigeration, which is what happened in the cases in our study [22]. The sources of foodborne botulism in our study were mostly pork ham and canned meat. Besides the bacterial infections mentioned above, the hazards of consuming pork products include viruses, such as hog cholera, African swine fever, foot and mouth disease, and swine vesicular disease [23].

Home-preserved food is put in bottles or cans, and botulism occurs because of a combination of inadequate preparation and improper storage. Pork containers are treated in a running water reservoir and the contents were eaten without rewarming. Additionally, this study found four cases from the consumption of preserved pork and pâté. Home-preservation of food is common in rural areas because it is economical and there is widespread accessibility of uncooked food [24].

Among geographic regions, the type of botulinum toxin varies, indicating altered food inclinations and accessibility. In the U.S. and Asia, outbreaks are mainly due to toxin type A, type B in Europe, and type E in Canada [22]. Type B toxin was associated with 79.1% of confirmed cases in Italy followed by type A [3]. Globally, type A was found in 34% of epidemics, followed by type E (17%) and type B (16%). Type B was isolated in all the cases in our study; no type A or E was detected. The study linked to the epidemiology of botulism eruption discovered that the incubation phase was considerably shorter in type E epidemics compared with type A or B. The mean ratio of events that needed ventilatory support was more common in type A epidemics than for type B [15]. 

This study had certain limitations as it was a retrospective study, and data were extracted from clinical records. In addition, for most patients the presence of a BoNT was laboratory-confirmed, but there could have been misclassification in patients with dual toxin-producing strains of *Clostridium botulinum,* where the dominant one caused the other to go unnoticed during botulism investigation [25]. The toxin also might not have been spotted because its levels were below the detection limit or an alteration took place during transport. In some epidemics, the level of toxin below the detection limit occured in 30–40% of cases [26]. The trivalent anti-botulinum toxin caused allergic reactions (i.e., rash, erythema) in seven patients (38.9%), but they made a complete recovery. In a different study [27], among 687 patients treated with the heptavalent serum, 45 (6.5%) developed allergic reactions, of which eight (1.2%) were anaphylactic events The patients needed antihistamine and epinephrine management and, ultimately, intensive patient care. Another limitation of the current study was the lack of examination on food portions; thus, the food source could not be confirmed.

## 5. Conclusions

Although an unusual disease, botulism should be included in a diagnosis of individuals with neurological symptoms. Botulism related to contaminated food has been the most common form in western Romania throughout the previous decade. In this study, mostly home-preserved pork products contributed to several botulism incidents. Probing the occurrence of foodborne botulism offers valuable information concerning food and the circumstances that promote toxin formation. Botulism may be averted in high-occurrence districts by recognizing cooking and community standards and discouraging local traditions to prevent botulism. Stronger attempts should be made to educate the public on botulism mindfulness and its associated consequences, and deterrence-centered food safety regulators can aid in preventing epidemics.

## Figures and Tables

**Figure 1 healthcare-09-01149-f001:**
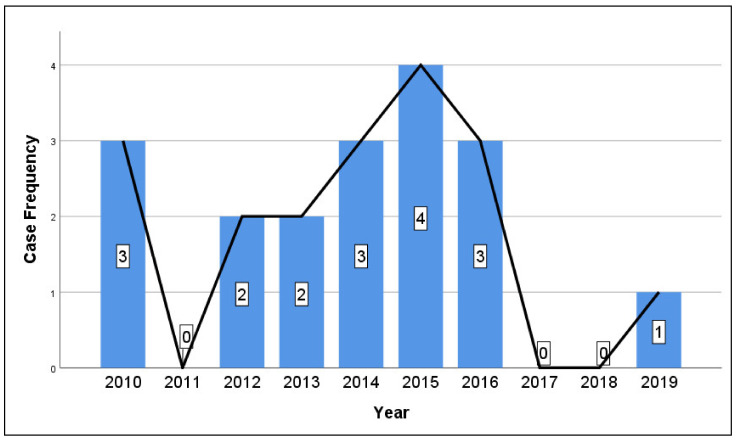
Yearly case frequency of foodborne botulism in Western Romania.

**Figure 2 healthcare-09-01149-f002:**
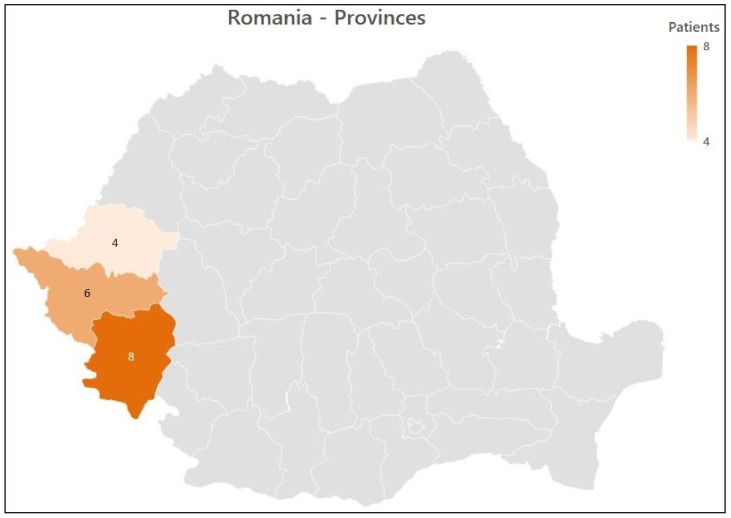
Case distribution by province.

**Table 1 healthcare-09-01149-t001:** Presenting symptoms of patients with foodborne botulism.

Symptoms	Patients	Frequency
Dysphagia	18	100%
Headache	16	88%
Dizziness	15	83%
Double Vision	12	66%
Constipation	12	66%
Ptosis	12	66%
Fatigue	11	61%
Insomnia	11	61%

**Table 2 healthcare-09-01149-t002:** Demographic characteristics of patients admitted with botulism.

Characteristic	Number (Proportion)
Age	48.4 ± 17.5
Men	11 (61.1%)
Complications	1 (5.6%)
Urban Rural	4 (22.2%) 14 (77.8%)
Allergic Reaction to Antitoxin	7 (38.9%)
Co-existing Medical Condition	6 (33.3%)

## Data Availability

Data supporting reported results can be found at https://www.cnscbt.ro/index.php/rapoarte-anuale accessed on 26 January 2021.

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
