# Peer review of "Foodborne Botulism in Western Romania: Ten Years’ Experience at a Tertiary Infectious Disease Hospital"

_healthcare, 2021, doi:10.3390/healthcare9091149_

Round 1
Reviewer 1 Report
The manuscript is definetly of interest to be published and should be accepted on the condition that the following concerns are addressed:
please consider revising the keywords
L36-37: please rephrase the sentence, as it is grammatically incorrect
L40-41: please include the following reference:
https://www.ncbi.nlm.nih.gov/pmc/articles/PMC7391379/
L51-54: please rephrase the sentence and discuss it in more detail and more accurately.
L60: Because these symptoms..
L65: weeks to months, which is influenced by…
Please provide more information on the study site (L82)
Table 2. Demographic characteristics
You dont need to identify males and females, one is enough
L173-183: please discuss treatments in more detail
L184-199: please include and discuss the following references:
https://www.hindawi.com/journals/jfq/2019/1048092/
https://pubmed.ncbi.nlm.nih.gov/9329109/
https://pubmed.ncbi.nlm.nih.gov/32847011/
L220-225: please rephrase and improve grammar
Author Response
Dear editorial team,
Thank you for considering our manuscript, as we appreciate your efforts to analyze and improve our paper. Thus, based on the feedback received during the review phase, our team had carefully revised the article that also underwent extensive English editing, with the following changes:
- Line 36-37 was paraphrased
- Line 40-41: included the following reference https://www.ncbi.nlm.nih.gov/pmc/articles/PMC7391379/
- Line 51-54: the sentence was rephrased to a more accurate description
- Line 60: Because these symptoms are not disease-specific, the changes in appearance can make diagnosis challenging.
- Line 65: Complete recovery usually takes weeks to months, which is influenced by prompt administration of heptavalent botulinum antitoxin serum and respiratory support when needed.
- Line 82: We provided more information on the study site and dates.
- Line 98: The sentence was rephrased “to establish the botulinum toxin subtype”
- The captions of tables and figures were extended.
- Table 2: the row with women was removed
- Line 105: The sentence was rephrased to “The majority of patients exhibited gastrointestinal symptoms…”
- Line 117: we added “comorbid”
- Line 154: The sentence was rephrased
- Line 173: We discussed treatment options in more detail.
- Line 184: We included and discussed the following references:
https://www.hindawi.com/journals/jfq/2019/1048092/
https://pubmed.ncbi.nlm.nih.gov/9329109/
https://pubmed.ncbi.nlm.nih.gov/32847011/
- Line 188: the sentence was rephrased to “It has formerly been observed that C. botulinum neurotoxin types B, E, and F can grow at low temperatures used for food refrigeration, as what happened in the cases presented in our study”
- Line 194: we added "patients with botulism"
- Line 201: we used "outbreaks" instead of "outbursts"
- Line 207: we replaced “competed to” with “competed with”
- Line 213: Clostridium botulinum was changed to italics
- Line 220: We rephrased and improved the grammar.
Best regards,
The authors

Reviewer 2 Report
The study by Marincu et al. aimed to analyze the epidemiological data on foodborne botulism in Western Romania in the last decade. In the period, the authors evidenced 18 confirmed cases of botulism, most of which had the consumption of pork ham and canned meat as a source. The authors described the clinical manifestations of the patients and associated all cases with toxin B. The text is somewhat confusing in some parts, in which it would need to be rewritten.
Overall comments:
- Line 98: Please check whether you meant “botulism” instead of “botulinum” in the end of the sentence
- In results: The captions of tables and figures are very summarized. The authors could describe each one a little bit.
- Line 105: Please clarify what you mean by "higher patients".
- Line 117: Please use comorbid instead of co-morbid
- Line 154 – 158: The sentence that starts with "It is expected..." to "...food preservation" is too confusing. Please the authors should make it clearer.
- Lines 188-189: this sentence seems incorrect. Please clarify what you mean by "C. botulinum neurotoxin types B, E and F can grow in chilling temperatures". Can the microorganism grow at these temperatures? Are neurotoxins more stable at these temperatures? Or can they be produced at these conditions?
- Line 194: I suggest the authors add "with botulism" after "patients" (patients with botulism in our study)
- Line 201: I suggest the authors use "outbreaks" instead of "outbursts"
- Line 207: I suggest the authors use “competing with” instead of “competed to”
- Line 213: Clostridium botulinum should be in italics
Author Response

(The authors gave the same response as above.)

Round 2
Reviewer 2 Report
The authors accepted all suggestions. The manuscript is now acceptable for publication.
This manuscript is a resubmission of an earlier submission. The following is a list of the peer review reports and author responses from that submission.
Round 1
Reviewer 1 Report
This paper reports on 18 probable foodborne botulism cases among patients admitted to a hospital in West Romania during 10 years, as deduced by a retrospective analysis of clinical records.
The data could provide a contribution to knowledge of the geographic distribution of foodborne botulism cases in the last 10 years.
However, the manuscript in its current form has several important conceptual and experimental limitations and inaccuracies which greatly limit its potential usefulness.
First, many definitions of importance in an epidemiological study are lacking, including the case definition of foodborne botulism (probable or confirmed), as well as the definitions of outbreaks and epidemics adopted by the authors.
Moreover, the authors state that “the alleged food portions were not examined”: hence, no food source was confirmed, although for 14 cases there was a possible link with the consumption of conserved pork products. No details are provided for the remaining 4 cases: how the authors exclude that those patients were affected by other botulism forms, e.g. wound botulism or infectious botulism?
Finally, the experimental procedure used to detect the botulinum neurotoxin in the serum samples from patients is only vaguely described, with no references cited. It is stated the toxin was identified on the first day of admission: however, results from the traditional mouse intraperitoneal biological test can be obtained after 3/four days from inoculation.
Author Response
Dear reviewer,
Thank you for considering our manuscript, as we appreciate your efforts to analyse and improve our paper. Thus, based on the feedback received during the review phase, our team had carefully revised the article with the following changes.
Line 10: The word “infection” was replaced with “illness”. The same edit was done to all other cases.
Line 12: Clostridium Botulinum was written in italics, as well as for all other cases.
Line 29: “lethal infection” and country-specific data was modified based on given advice.
Line 35: We have modified the third and fourth sentences based on your recommendation and citation.
Line 36:
- “C. botulism” was changed to “C. botulinum”.
- Reference number 8 was replaced with your suggestion.
- botulinum “strains” was replaced with “neurotoxins”
Line 41: “food contaminated by BoNT” replaced the initial sentence.
Line 55: We have edited 4 sentences following line 55.
Line 60 was removed
Line 66: We have modified the sentences using “outbreak”, to properly describe and define the term.
Line 87: The BoNT was identified from serum samples taken on the first day of hospital admission, so, the toxin was not identified on the first day of admission.
Materials and Methods – Line 92: the AOAC official method was cited as the method used to identify the BoNT
Line 137: There was a mistake in reporting the food products involved in causing the disease to the 18 cases involved. All 18 cases were described.
Line 253: We have mentioned the lack of examination on the alleged food portions as a limitation of the current study.
Best regards,
The authors
Reviewer 2 Report
The reviewed paper regards to epidemiological records of botulism cases in Western Romania, during 10 years period since 2010 to 2020. The authors presented clinical observations among affected patients proved by MBA (mouse bioassay) results with seroneutralization.
The manuscript has a character of commucation about 20 botulism cases with detail record of clinical symptoms. Laboratory testing was limited only to MBA preparation for serum analysis. However, mentioned detail description of symptoms, toxin type determination, description of potential sources of BoNT and epidemiological record of botulism cases in Wertern Romania could be interesting to the readears, especially epidemiologists exploring foodborne botulism cases.
I would like to suggest some changes in the text, listed below, which should be introduced before publication:
Major notices:
Line 10: Please change the phrase "botulism is an unusual infection...". Generally, botulism is not infection, as you noticed in further part of your study, foodborne variant of this disease is caused by botulinum toxin, not by C. botulinum spores or cells. Foodborne botulism has a character of intoxication. Very rare cases of adult intestinal toxemia (also known as adult intestinal colonization) botulism can happen if the spores of the bacteria get into an adult’s intestines, grow, and produce the toxin. Toxicoinfection is also noticed in infant botulism and wound botulism. Please, see CDC classification: https://www.cdc.gov/botulism/definition.html
Line 12: Please, use an italics for description of species name in all the text of your study.
Line 29: Please change "lethal infection". I'm not agree with statement that botulism has advanced prevalence in Poland. Historically, prevalence of botulism was much more higher in Poland (tousends cases in 80s), especially in time before "economic transformation" that had been before 90s. Now, e.g. in 2018 - only 22 cases were noted and in 2019 -15 cases (morbidity 0.04-0.06/1000 people). It is not "advanced prevalence". See annual reports of the The National Institute of Public Health – National Institute of Hygiene or NIPH–​NIH in Poland. Please, verify this information. I agree that prevalence was advanced in the past.
Line 35: The first accurate and complete description of the clinical symptoms of food-borne botulism was published between 1817 and 1822 by the German physician and poet Justinus Kerner (1786–1862), who also developed the idea of a possible therapeutic use of botulinum toxin, which he called “sausage poison-botulus”. It was 19th century. Please, verify this information and see: Erbguth FJ, Naumann M. Historical aspects of botulinum toxin: Justinus Kerner (1786-1862) and the "sausage poison". Neurology. 1999 Nov 10;53(8):1850-3. doi: 10.1212/wnl.53.8.1850. PMID: 10563638.
Line 36: Please correct: C. botulism into C. botulinum. Now, we differentiate 8 botulinum neurotoxins. Botulinum neurotoxins (not strains of C. botulinum) are known to have eight serotypes (BoNT/A-G and X). Botulinum toxins are produced not only by Clostridium botulinum - see: Michel R. Popoff, Botulinum Neurotoxins: Still a Privilege of Clostridia?, Cell Host & Microbe, Volume 23, Issue 2, 2018, Pages 145-146.
Line 41: Please change this phrase: "After consuming unclean food...". I think that food contaminated by BoNT sounds better.
Line 55: 12-15 days of incubation is characteristic for toxicoinfection, I suggest to remove this phrase
Line 60: "...isolating Clostridium botulinum in stool or foodstuff ...is inconvenient..." - please, define why? I suggest to add one sentence about heterogeneity of BoNT -poducing clostridia (one of the reasons with isolation) or simply remove this phrase.
Line 76: Materials and Methods - please describe Mouse Lethality Assay in more details or cite e.g. AOAC procedure for BoNT detection in serum.
Despite of some notices listed above, In my opinion this manuscript is valuable study.
Kind regards,
One of the reviewers
Author Response

(The authors gave the same response as above.)

Round 2
Reviewer 2 Report
Dear Authors,
thank you for the introducing my suggestion. In my opinion, the manuscript is thoroughly improved. Hovewer, I have one additional notice:
line 40: I would like to suggest to remove the information about toxin type H, because it was proved by Maslanka et al. (2015) that BoNT/H is not separate toxin and it has a hybrid-like structure containing regions of similarity to the structures of BoNT/A1 and BoNT/F5. I would like to suggest to add info about toxin type X described by Zhang et al. (2017).
[Zhang S, Masuyer G, Zhang J, Shen Y, Lundin D, Henriksson L, Miyashita SI, Martínez-Carranza M, Dong M, Stenmark P. Identification and characterization of a novel botulinum neurotoxin. Nat Commun. 2017 Aug 3;8:14130. doi: 10.1038/ncomms14130. PMID: 28770820; PMCID: PMC5543303.]
I suggest to rewrite line mentioned above by replacing H (controvertial) with X.
This is only minor suggestion. I recommend your study for publication.
Kind regards,
One of the reviewers
Author Response
Dear reviewer,
Thank you once again for all the offered support!
Based on your recommendation, we have modified line 40 to include the BoNT type X, as well as we have included the articles by Maslanka et al. (2015) and Zhang et al. (2017) in the Bibliography (numbers 8 and 9).
Best regards,
The authors
